



# Centuries of intense surface melt on Larsen C Ice Shelf

Suzanne Bevan[1], Adrian Luckman[1], Bryn Hubbard[2], Bernd Kulessa[1], David Ashmore[3], Peter Kuipers Munneke[4], Martin O'Leary[1], Adam Booth[5], Heidi Sevestre[6], and Daniel McGrath[7,8]

[1]Geography Department, College of Science, Swansea University, Singleton Park, Swansea, SA2 8PP, UK
[2]Centre for Glaciology, Department of Geography and Earth Sciences, Aberystwyth University, Aberystwyth, UK
[3]School of Environmental Science, Roxby Building, University of Liverpool, L69 7ZT, UK
[4]Institute for Marine and Atmospheric Research, Utrecht (IMAU), Utrecht University, P.O. Box 80000, 3508 TA Utrecht, The Netherlands
[5]School of Earth and Environment, University of Leeds, Leeds, LS2 9JT, UK
[6]School of Geography and Geosciences, University of St Andrews, College Gate, St Andrews, KY16 9AJ, UK
[7]Department of Geosciences, Colorado State University, Colorado, USA
[8]U.S. Geological Survey, Alaska Science Center, Anchorage, AK, USA

*Correspondence to:* S. L. Bevan (s.l.bevan@swansea.ac.uk)

**Abstract.**

Following a southward progression of ice-shelf disintegration along the Antarctic Peninsula, Larsen C Ice Shelf is the focus of ongoing investigation regarding its future stability. The ice shelf is known to be experience surface melt, and commonly features surface meltwater ponds. Here, we use a flowline model and a firn density model to date and interpret observations of melt-affected ice layers found within five 90 m boreholes distributed across the ice shelf. We find that units of ice within the boreholes, which have densities exceeding those expected under normal compaction metamorphism, correspond to two climatic warm periods within the last 300 years on the Antarctic Peninsula. The more recent warm period, from the 1960s onwards, has generated distinct sections of dense ice in two boreholes in Cabinet Inlet, close to the Antarctic Peninsula mountains — a region currently affected by föhn winds. Previous work has classified these layers as refrozen pond ice, requiring large quantities of mobile liquid water to form. Our flowline model shows that, whilst preconditioning of the ice began in the late 1960s, it was probably not until the early 1990s that twentieth-century ponding began. The earlier warm period occurred during the 18th century and resulted in two additional sections of anomalously dense ice deep within the boreholes. The first, in one of the Cabinet Inlet boreholes, consists of ice characteristic of refrozen ponds and must have formed in an area currently featuring ponding. The second, in a mid-shelf borehole, formed at the same time in an area which now experiences significant annual melt. Further south on the shelf, the boreholes sample ice that is of an equivalent age but which does not exhibit the same degree of melt influence. This west–east and north–south gradient in past melt distribution resembles current spatial patterns of surface melt intensity. Using flowlines to trace the advection and submergence of continental ice identified in boreholes, we demonstrate that, even by the time the ice reaches the calving front, only the upper 40 to 50% of the shelf is composed of meteoric ice accumulated on the shelf. This vertical composition implies that basal crevasses must be confined within continental and/or basally accreted ice, and therefore will be unaffected by current climate-induced firn compaction.



# 1 Introduction

With an area of ~52,000 km$^2$, Larsen C Ice Shelf (LCIS) on the eastern Antarctic Peninsula (AP) is the fourth largest ice shelf in Antarctica. Following the southward progression of ice-shelf disintegration along the AP since the 1950s, including the loss of Prince Gustav and Larsen A in 1995 (Rott et al., 1996) and Larsen B in 2002 (Rott et al., 2002), the stability of LCIS would seem to be at risk. In 2000, LCIS was closely bounded to the north and west by the -9°C surface mean annual isotherm considered to be the northerly limit of ice-shelf viability (Morris and Vaughan, 2003). In recent decades at least, LCIS, and particularly its northern sector, has exhibited a number of factors that are indicative of instability: surface lowering (Shepherd et al., 2003; Fricker and Padman, 2012; Holland et al., 2015) at an increasing rate (Paolo et al., 2015), firn air depletion (Holland et al., 2011), recession (Cook and Vaughan, 2010), and surface ponding (Luckman et al., 2014). In 2014 a rift began to propagate from the south which is likely to cause ~ 10 % of the ice shelf area to break away in the near future: the largest calving event since the 1980s (Jansen et al., 2015; Borstad et al., 2017).

Surface lowering and ice-shelf thinning on LCIS is a result of both firn air depletion and ice loss (Holland et al., 2015). The ice loss may be a result of reductions in basal ice accretion, or increases in basal melt or flow divergence; firn air may become depleted because of reductions in accumulation or because of enhanced surface melt and refreezing within the firn. The areas on LCIS with low firn air (Holland et al., 2011), to the north and in the lee of the mountains, coincide with areas where annual melt duration is longest and where föhn winds influence the surface (Luckman et al., 2014). A föhn wind is an orographic phenomenon that, on LCIS, occurs under moderate to strong westerly flow and leads to warm and dry air displacing the prevailing cool near-surface conditions (Elvidge et al., 2015).

Recent trends in surface melt parameters on the Antarctic Peninsula, such as onset date and duration or intensity, depend on the time period under consideration. During the second half of the 20th century, mean annual temperatures were increasing (Vaughan et al., 2003) and melt trends, based on sums of positive degree days (PDDs) from the few meteorological stations with measurements during this period, showed increases significant at the 95% confidence level or better (Vaughan, 2006). When analyses include the first decade of the 21st century, during which time AP mean annual temperatures decreased (Turner et al., 2016), the trends based on PDDs remained positive but less steep (Barrand et al., 2013). The various studies do however reveal a large amount of interannual variability, with annual meltwater volume varying by a factor of 4 during the period 1979 to 2010 (Kuipers Munneke et al., 2012).

During years in which in which surface melt periods are long and intense, widespread ponds may form on ice shelves. Such ponds have been proposed as a trigger for ice-shelf break-up, either by enhancing the hydrofracture of existing crevasses (Scambos et al., 2000; MacAyeal et al., 2003; van den Broeke, 2005; McGrath et al., 2012), or via the stresses induced by hydrostatic rebound following drainage (MacAyeal and Sergienko, 2013), which may lead to runaway disintegration (Banwell and Macayeal, 2015). Spatially extensive surface ponding occurred on both Larsen A and B prior to break-up (Sergienko and Macayeal, 2005). Such extensive ponding has yet to be observed on LCIS but Holland et al. (2011) highlighted Cabinet Inlet on the Foyn Coast of LCIS (Fig. 1) as a particular location where observations in optical satellite images of surface melt ponding coincide with low firn air content, and with low backscatter in summertime synthetic aperture radar (SAR) images,



indicative of high surface water content. A recent study based on borehole images and profiles of temperature and density, ground-penetrating radar, firn-density modelling, and satellite images identified a massive subsurface body of anomalously warm and dense ice in Cabinet Inlet (Hubbard et al., 2016). This body of ice was interpreted to be the result of the intense melt and regular surface ponding that occurs in this area.

The effect of föhn-induced melting and ponding on the englacial properties of the ice shelf downstream depends on the history of surface melt over the last 600 to 800 hundred years, and consequently how far along-flow its temperature and density legacy has been advected. Almost all surface meltwater that percolates down through the snow and firn, refreezes in the firn and releases latent heat, thereby increasing the density and the temperature of the subsurface layers (Vaughan, 2008) and changing the rheology of the ice shelf. The increase in temperature reduces the viscosity of the ice allowing an acceleration of

ice flow relative to colder ice, potentially increasing lateral rifting and leading to the possibility of ice shelf break-up (Rack and Rott, 2004). The increase in density may, however, compensate for the temperature effect by increasing the fracture toughness and stabilising the ice against crevassing (Rist et al., 2002; Jansen et al., 2010). Without direct observations of density and temperature, modelling experiments designed to investigate the stability of the ice shelf must tune rheological parameters to minimise the misfit between modelled and observed velocity fields (e.g., Vieli et al., 2007; Furst et al., 2016).

In this research we investigate where and when various units of melt-affected ice observed within the Cabinet Inlet and four other boreholes originated, relate the origins to past local climate, and estimate how much of the ice shelf is likely to be affected.

The additional boreholes consist of two directly downstream from Cabinet Inlet, and two downstream from Whirlwind Inlet (Fig. 1). The 90-m boreholes were drilled and viewed using an optical televiewer (OPTV) at site CI-0 in November 2014, and

at the four other sites, CI-22, CI-120, WI-0 and WI-70, in November and December 2015 (Fig. 1). The OPTV borehole images, with a pixel size of 1 mm$^2$, provide information about the material composition and structure of the borehole walls (Hubbard et al., 2008). Firn density may be derived from the luminosity of the recorded image (Hubbard et al., 2016), and relies on an empirical relationship between image brightness and density; with lower reflectivity corresponding to denser ice (Hubbard et al., 2013).

The five borehole images are described in detail in Hubbard et al. (2016) and Ashmore et al. (2017) (Figs. 2 and 3 reproduced from Ashmore et al. (2017)). Across the sites, four different ice types, or units, are identified on the basis of visual appearance, density, and refrozen ice content. Image thresholding was used to determine the proportion of ice within each unit that is composed of refrozen infiltration ice. Unit 1 (U1) is the uppermost unit and is the only unit observed in all boreholes: it is interpreted as accumulated snow or firn undergoing compaction metamorphism with sporadic but spatially widespread layers

formed by melt–refreeze events. Unit 2 (U2) is present only at depth at site CI-120 and is composed of ice that has experienced enhanced compaction owing to higher temperatures and surface melt; the host ice is dense but refrozen layers are still visible within the column. Unit 3 (U3) occurs only at CI-0 and CI-22 and is interpreted as refrozen pond ice (Hubbard et al., 2016). The ice in U3 is homogeneous with only diffuse layering. Extreme melt events are required to allow a sufficient quantity of mobile melt-water to percolate down and add to the previous upper surface of U3. Finally, Unit 4 (U4) is the least dense of all




the units, containing steeply dipping layers of deformed ice, and is identified as continental ice originating from upstream of the grounding line. U4 is present at the base of the CI-0 and WI-0 boreholes.

The significant quantities of refrozen ice within the boreholes suggests that intense melt is spatially pervasive and has been ongoing on LCIS for decades or even centuries (Ashmore et al., 2017). We use a flowline model based on measured surface
velocities and modelled surface mass balance (SMB) to determine the spatial pattern and history of the melt.

## 2 Data and methods

### 2.1 flowline model

Following Craven et al. (2009) and McGrath et al. (2014), a flowline model was constructed for LCIS which simulates the advection and submergence of surface layers along trajectories passing through the borehole sites. We define trajectory as the
surface route based on velocity information only. From any specified starting point, surface trajectories were created which allow the path length through, and hence time spent in each grid cell to be determined.

Following any trajectory, the rate of change of thickness of a surface layer $Z$ is given by

$$\frac{DZ}{Dt} = \frac{\partial Z}{\partial t} + \boldsymbol{u}\frac{\partial Z}{\partial x} = Z\dot{\epsilon}_z + \dot{a} + \boldsymbol{u}\frac{Z}{H}\frac{\partial H}{\partial x} \tag{1}$$

where $\boldsymbol{u}$ is the along-flow velocity, $\dot{\epsilon}_z$ is the vertical strain rate, $\dot{a}$ is the surface mass flux rate, and $H$ is the ice-shelf
thickness, sampled from Bedmap2 (Fretwell et al., 2013). Equation (1) allows us to determine the three-dimensional path from the surface and through the ice shelf of an ice particle, we will refer to this path as a flowline.

Surface trajectories were generated that passed directly through CI-0 and through WI-0, and subsequently within 1 km of each of the downstream borehole sites. The upstream and downstream limits of these trajectories were determined by the spatial extent of the velocity data. Next a series of flowlines were initiated at selected points along the trajectories so that from these
points the accumulated snow and ice thicknesses match the depths of the interfaces between the different units at each of the borehole sites.

### 2.2 Velocity data

The trajectory routing was based on flow vectors from the 450 m version of the NASA Making Earth System Data Records for Use in Research Environments (MEaSURES) Antarctic velocity dataset (Rignot et al., 2011c, b). The same velocity dataset,
smoothed with a low-pass 4.5 km Gaussian filter, was used to calculate the strain rate along each trajectory. In order to estimate the sensitivity of our results to velocity we recomputed the along-flow rates of accumulation and submergence using $\pm 10\%$ velocity magnitudes (Figs. 4a and b). This percentage change is a conservative estimate of velocity error which was quoted to be a maximum of 17 m/yr (Rignot et al., 2011b).



## 2.3 Surface mass flux

Surface mass fluxes were computed using RACMO2, a regional atmospheric climate model adapted for simulations of polar climate. Annual means of surface mass balance ($\dot{a}$) for the period 1979–2014 were calculated for a domain covering the AP and surrounding seas at a horizontal resolution of approximately 5.5 by 5.5 km (van Wessem et al., 2015). The surface mass flux rate along each profile, $\dot{a}$ in Eq. (1), was based on the median rate, with errors based on upper and lower quartiles, over the 1979–2014 period (Figs. 4a and b).

We converted surface mass fluxes to thickness using density derived from our borehole profiles (Ashmore et al., 2017) from borehole CI-120 along flowlines from Cabinet Inlet, and WI-70 along flowlines from Whirlwind Inlet. CI-120 densities range from less than 700 kg m$^{-3}$ at a depth of 2 m to 903 kg m$^{-3}$ at a depth of 90 m (Fig. 2c). WI-70 densities are less than 600 kg m$^{-3}$ at a depth of $\sim$2 m and $\sim$900 kg m$^{-3}$ at 90 m (Fig. 3b). At each step along the flowline, after accumulating the appropriate amount of surface mass, the density was adjusted to the depth-mean density appropriate to the total thickness accumulated up to that point. In this way we modelled the natural compression of accumulated firn as it was advected downstream.

## 2.4 Firn density model

In order to predict the time evolution of the near-surface firn density profile at a single location within Cabinet Inlet we ran a one-dimensional firn densification and hydrology model (FDM) (Ligtenberg et al., 2011). The model is driven by mass fluxes, wind speed and surface temperature from RACMO2 and takes into account firn compaction, meltwater percolation and refreezing.

## 3 Results

Each flowline, triggered from a point along each of the Cabinet and Whirlwind Inlet trajectories, allows an age to be estimated for the transition or interface between the units observed in the borehole images, and also for the bases of the boreholes (Table 1). The flowlines also pinpoint where on the ice shelf surface the transition between units originated (Figs. 5a and b). Uncertainties in ages and distances travelled may be a result of measurement (velocity, density) or model error (SMB). They may also result from conditions having changed through time, although Glasser et al. (2009) argued that the persistence of surface features down-flow suggests minimal change in flow speed and direction over at least the last 560 years. The age ranges given in parentheses in Table 1 and the shaded bounds in Figs. 5a and b are the result of using the lower quartile of SMB combined with +110% velocity magnitude, and the upper quartile of SMB combined with 90% velocity magnitude.

No ages are calculated for the bases of CI-0 and WI-0 as they originate upstream of the available velocity data. From the start of the flowline to the edge of the shelf each trajectory is about 200 km in length, and it takes 870 and 588 years for the ice to be transported from Cabinet Inlet and from Whirlwind Inlet, respectively (Figs. 5a and b).





**Table 1.** Depths of the unit interfaces observed in borehole images (Ashmore et al., 2017), and age of surface origin based on the flowline modelling.

| Borehole | Unit interface | Depth (m) | Age (years) | |
|---|---|---|---|---|
| CI-0 | U1/U3 | 2.9 | (not resolvable) | |
| | U3/U4 | 44.87 | 118 | (86–127) |
| | Base | 97.50 | - | |
| CI-22 | U1/U3 | 5.9 | 17 | (15–18) |
| | U3/U1 | 15.64 | 47 | (40–54) |
| | U1/U3 | 61.68 | 204 | 168–236) |
| | Base | 90.0 | 266 | (246–268) |
| CI-120 | U1/U2 | 68.56 | 281 | (229–331) |
| | Base | 90.0 | 433 | (279–559) |
| WI-0 | U1/U4 | 64.95 | 84 | (72–92) |
| | Base | 90.0 | - | |
| WI-70 | Base | 90.0 | 227 | (186–299) |

The time-dependent FDM output for CI-0 from steady state in 1979 until 2014, shows the variation in firn accumulation over glacier ice (density 917 kg m$^{-3}$) driven by the ratio of melt to accumulation processes (Fig. 6). Throughout the time series the firn is interspersed with high density layers caused by intermittent melt events. A major melt event in 1993 removed all firn, and the subsequent years of high melt combined with low accumulation left the dense ice within a meter of the surface until 2009. By 2014, as reported in Hubbard et al. (2016), a 2.9 m layer of firn had re-established consistent with the borehole observations.

## 4   Interpretation and discussion

Interfaces between the ice units identified in the borehole logs, when traced back to the surface using the flowline model, may be interpreted in terms of either spatial or temporal changes in surface melt. For example, a transition from U1 (a unit with only sporadic melt layers) down into U2 (a unit with a high proportion of melt–refreeze) within a borehole, might be a result of ice flow having passed from a region of high to low melt conditions, or a result of a temporal switch from high to low surface melt conditions at the time the ice in transition was at the surface. In discussing the transitions we refer to the upper/younger unit first and then the lower/older unit, for example, the boundary between U3 and U4 at CI-0 is referred to as U3/U4.



## 4.1 Continental ice, U4

In Cabinet Inlet the origin of the continental ice, marked by the U3/U4 transition, can be traced back 22 km upstream of CI-0 which equates to an advection time of 118 years (Fig. 5a). The U3/U4 origin coincides with the grounding line based on visual inspection of MODIS imagery (Bohlander and Scambos, 2007) but is about 9 km upstream of the grounding line identified using differential satellite radar interferometry (Rignot et al., 2011a) which marks where the ice first goes afloat. For Whirlwind Inlet the corresponding transition, this time U1/U4 (Fig. 5b), takes place 12 km downstream of the both the MODIS and the interferometric grounding lines. This 12 km discrepancy may indicate grounding line retreat over the last 65 years or an overestimation by the model of SMB close to the base of the mountains. Modelled SMB is almost a factor of two higher at the grounding line than it is at WI-0 (Fig. 4b). Thus the flowline model confirms that U4 in both CI-0 and WI-0 borehole logs is continental ice, in line with Ashmore et al. (2017).

Continuing the flowline to trace the U3/U4 interface downstream from CI-0 predicts U4 to be at a depth of 67 m by the time it reaches CI-22. Ashmore et al. (2017) did not identify any continental ice within the 90 m borehole suggesting that the SMB may be underestimated; the lower depth based on error bounds puts U4 at 89 m at CI-22. Farther downstream from Cabinet and Whirlwind Inlets, the modelled depth of U4 is below the bases of both CI-120 and WI-70 boreholes and, in broad agreement with McGrath et al. (2014), we find that approaching the edge of the ice shelf, locally accumulated meteoric ice accounts for between 40 % and 50 % of the ice column. Luckman et al. (2012) observed and modelled basal crevasse penetration heights to be limited to ∼200 m; our modelling suggests that this is mostly below the depths of locally accumulated meteoric shelf ice. Therefore, if basal crevasses are limited to continental or basal accreted ice, firn compaction under climate warming may not be a factor which would contribute to increasing penetration depths.

## 4.2 Refrozen pond water, U3

U3 ice, which by its nature would have required sufficient surface melt to allow percolation and refreezing in continuous vertical units and probably surface ponding, is only observed in the Cabinet Inlet boreholes (Fig. 2). At CI-0, it lies beneath 2.9 m of U1 and extends to a depth of 44.87 m, and at CI-22 the upper section of U3 is covered by 5.9 m of U1. The lower section of U3 at CI-22 extends from 61.68 m to the base at 90 m.

CI-0 was logged in November 2014, and the 2.9 m of U1 had a 6 year period of accumulation with no evidence, either from firn density modelling or in satellite images (Hubbard et al., 2016; Ligtenberg et al., 2011), of ponding. We do observe ponding in MODIS imagery of Cabinet Inlet during early 2015, prior to CI-22 being logged in November 2015. However, CI-22 is ∼2 km downstream of the area of observed ponding. We can use the lower boundaries of the U3 sections to estimate the earliest date at which surface ponds could have been forming, although the potentially mobile nature of large volumes of meltwater means that the actual origins of U3 could have been much later in time, and farther downstream. For the U3/U4 interface at CI-0 (44.87 m) this date is 118 years before present (BP), and 47 years BP for the U3/U1 interface at CI-22 (15.74 m) (Table 1). The earliest date we can identify ponds in optical satellite imagery is in a Landsat 5 image for 02/02/1997; this does not mean that ponds were not present before this date, only that none have been observed in cloud-free images. The



FDM model (Fig. 6) suggests that 1993 was the first year, at least within the 1979–2015 interval, that the firn was capable of supporting ponds.

Climate reconstructions based on borehole temperatures on the Bruce Plateau (Zagorodnov et al., 2012) and station data from Orcadas, ∼700 km to the north-east of the tip of the AP (Zazulie et al., 2010), indicate that 20th century warming on the AP began in the 1950s. The Southern hemisphere Annular Mode (SAM) or Antarctic Oscillation index describes the difference in zonal-mean geopotential heights between mid and high latitudes which drives the strength and latitude of the sub-polar westerly winds. In the late 1960s the SAM entered a phase of increasing positive indices, particularly in summer and fall (Thompson and Solomon, 2002; Marshall, 2003), indicating a strengthening of the Antarctic circumpolar vortex, bringing strong westerly air flow to the AP. At meteorological stations to the north-east of the AP, positive SAM indices in summer and autumn are correlated with high air temperatures (Marshall et al., 2006). Further south, the mechanisms by which the associated strong westerlies are able to increase air temperatures over LCIS include the advection of warm air over the mountains (van den Broeke, 2005; Marshall et al., 2006), the blocking of cold southerly flow from the continent (Orr et al., 2004), and the enhancement of the föhn effect (Marshall et al., 2006; Elvidge et al., 2015). A period of intense surface melt in 2001/02, generated by unusually high frequencies of north-westerly winds, probably triggered the break-up of Larsen B ice shelf (van den Broeke, 2005). Föhn events on LCIS have become more common since the 1960s, are significantly ($\geq 98\%$) correlated with surface melt in the northern inlets of LCIS close to the base of the mountains (Cape et al., 2015), and have probably led to ponding in Cabinet Inlet (Luckman et al., 2014; Elvidge et al., 2015). We therefore propose, on the basis of the borehole evidence, the flowline model results for the U3/U1 interface at CI-22, and climatology, that the ponding which has led to the formation of the upper sections of U3 has become a feature of Cabinet Inlet only since the late 1960s and maybe not until the early 1990s.

Constraining the time period for recent ponding to no more than 50 years means that the 42 m of U3 at CI-0 must also have accumulated over a similar period — much less than the 118 years indicated by the model. Either we are underestimating the SMB flux in the model, or there is a significant amount of lateral influx of meltwater, or both, in the region upstream of CI-0. There is evidence that RACMO-2 both underestimates snowfall (Munneke et al., 2014) and summertime SMB (Kuipers Munneke et al., 2017), and the regions closest to the mountains and downstream of the dominant zonal wind component may be most affected by poor topographic resolution. Both the modelled depth of the U4 continental ice at CI-22, and the comparison of the U4 origin with the interferometric grounding line suggest SMB in Cabinet Inlet may be underestimated. In addition, modelled SMB close to the grounding line in Cabinet Inlet is less than 50% of that in Whirlwind Inlet (Figs. 4a and b). Lateral influx of meltwater is also a realistic possibility in this locality. By analogy with observations within the percolation zone of the Greenland Ice Sheet (Harper et al., 2012; Machguth et al., 2016), the formation of spatially discontinuous impermeable near-surface layers of ice following melt–refreeze events, would facilitate horizontal flow of meltwater along and across the troughs in which the melt ponds form. Vertical infiltration at the boundaries of the ice barriers would result in a horizontally heterogeneous distribution of U3 type bodies. As described in Hubbard et al. (2016), borehole CI-0 was drilled into a melt-pond trough which might be expected to contain a local concentration of infiltration ice.





The existence of a deep section of U3 at CI-22 between 61.68 m and the borehole base, indicates that surface ponds were also forming during an earlier period, which terminated 200 years BP. Deuterium content analysis of a 1000 year ice core from James Ross Island (JRI), 350 km to the north-east of Cabinet Inlet, shows mean annual temperatures to have been rising steadily for the past 600 years (Mulvaney et al., 2012; Abram et al., 2013) but that within this period 1777 AD marked the end

of one of two significant warming intervals. This warming interval repeatedly produced temperatures equivalent to those seen in the latter half of the 20th century, and may have conditioned the ice in Cabinet Inlet to a density at which surface ponding could occur, in a manner similar to that which has probably been occurring since the AP warming which began in the mid 1950s (Zagorodnov et al., 2012; Turner et al., 2016). In other words, the formation of a deep U3 section at CI-22 coincided with an anomalously warm period during the first half of the 18th century.

**4.3 Enhanced compaction unit, U2**

Unit 2 is only seen at the base of the borehole at CI-120 (Fig. 2). U2 is not as homogeneous as U3, but it does contain evidence of intense melt and densification exceeding that expected from compaction metamorphism alone (Ashmore et al., 2017). The top of U2 dates back to 281 (229–331) years BP (Table 1) and was possibly also affected by the 18th century warming discussed earlier in connection with U3. Tracing the top of U2 (the U1/U2 transition) places the down-flow boundary of the

region experiencing high melt at this time, to a point 77 km up-flow of CI-120. This origin is down-flow of the area currently affected by föhn winds (Luckman et al., 2014; Elvidge et al., 2015) which explains why U2 is not as dense and bubble free as U3 despite originating from the same period. The large borehole spacing does not allow us to draw any conclusions regarding the temporal persistence of this surface boundary in melt intensity. However, we know from contemporary observations of surface melt based on SAR and scatterometer data that the number of melt days per year, although highly variable, decreases

from west to east, that is down-flow, as well as from north to south across LCIS (Barrand et al., 2013; Luckman et al., 2014).

Along the trajectory originating in Whirlwind Inlet it is only as we reach WI-70 that we predict borehole sampling of ice as old as the U2 and U3 units at the bases of the boreholes at CI-22 and CI-120, respectively. That we do not sample any U2 ice at WI-70 is compatible with the clear north-to-south decrease in summer melt duration currently observed on LCIS (Barrand et al., 2013; Luckman et al., 2014).

**5 Conclusions**

We have used a flowline model to trace the surface origins of distinct units of ice observed in boreholes across LCIS. The units are characterized by varying amounts of refrozen ice content, and their spatial and temporal origins can be interpreted in the context of microclimate variations on the ice shelf, and AP climate change over the last 300 years.

From boreholes imaged along the Cabinet Inlet flowline we can deduce that warming from the mid 20th century precon-
ditioned the surface of the ice shelf within the inlet, until it became sufficiently impermeable to support surface ponds. The earliest possible date for 20th century pond formation, based on the 15.64 m base of U3 at CI-22, is the late 1960s. This date coincides with a switch to increasing positive SAM indices, a strengthening of the circumpolar vortex, and more frequent föhn




events. Firn density modelling starting from 1979 indicates that the surface was able to support ponds by 1993 and satellite observations confirm that ponds were present by the late 1990s.

Intense melt on the northern part of LCIS, and probably ponding within Cabinet Inlet, also occurred during the 18th century corresponding with an earlier period of warming over the AP. Unit 3 ice, at the base of the CI-22 borehole was forming up until 200 years BP, and U2 at the base of the CI-120 borehole up until about 280 years BP. Whilst U3 probably reflects the influence of föhn winds, U2 indicates more widespread warming over the shelf.

The pattern of melt reflected in the borehole logs, including the absence of U2 and U3 down-flow of Whirlwind Inlet suggests that past as well as recent melt is reflected in the current spatial distribution of firn air content (Holland et al., 2011).

By tracing the submergence of continental ice down-flow, we estimate that even by the shelf edge, where the proportion of shelf ice consisting of meteoric ice is at a maximum, only the upper 40 or 50% consists of local accumulation. Below this depth the shelf ice must consist either of continental ice or accreted marine ice. This vertical heterogeneity has implications for determining the resistance of the shelf to rifting and calving, it being the meteoric ice that exhibits the least resistance to rifting (McGrath et al., 2014). Basal crevasses are observed to penetrate upward through only the lower 200 m of shelf ice and therefore will be restricted to continental or basal accreted ice and not be affected by atmospheric processes acting on the firn.

This research demonstrates that the current setting of LCIS, featuring extensive, and in places intense, surface melt, and located immediately south of the boundary of AP ice-shelf viability, is not without precedence in the last 300 years. The previous AP warm period in the 18th century is captured in the stratigraphy of the shelf and is further evidence of the link between atmospheric warming and the collapse or decay of eastern AP ice shelves throughout the Holocene.

## 6 Data availability

The UK Polar Data Centre holds the flowline model code (doi:10.52850cea12bf-2f44-4d48-99d1-e7d303c5e80e) and results (doi:10.5285d363ff21-1576-4ad6-a2e8-bbc3c0a39b06). The MEaSURES Antarctic velocity dataset is available from the NSIDC (https:nsidc.orgdatadocsmeasuresnsidc0484_rignot) and the Bedmap2 dataset from British Antarctic Survey (https:secure.antarctica.ac.ukdata/bedmap2).

*Author contributions.* Suzanne Bevan carried out the flowline modelling and prepared the manuscript. Adrian Luckman led the project, Bryn Hubbard and David Ashmore supplied the borehole data, Martin O'Leary advised on the flowline modelling and Peter Kuipers Munneke supplied the SMB data. All authors contributed to drafting of the manuscript.

*Competing interests.* The authors declare that they have no conflict of interest.

*Acknowledgements.* The research was funded by the Natural Environment Research Council (NERC) grants NE/L006707/1 and NE/L005409/1. Navigation in the field was via a Leica VIVA GS10 GNSS loaned by the NERC Geophysical Equipment Facility loan number 1028. The

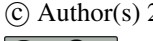



British Antarctic Survey provided logistical support in the field, and we thank our field assistants Ashley Fusiarski, Nick Gillett, Alan Davies and Bradley Morrell. Any use of trade, firm, or product names is for descriptive purposes only and does not imply endorsement by the U.S. Government.



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





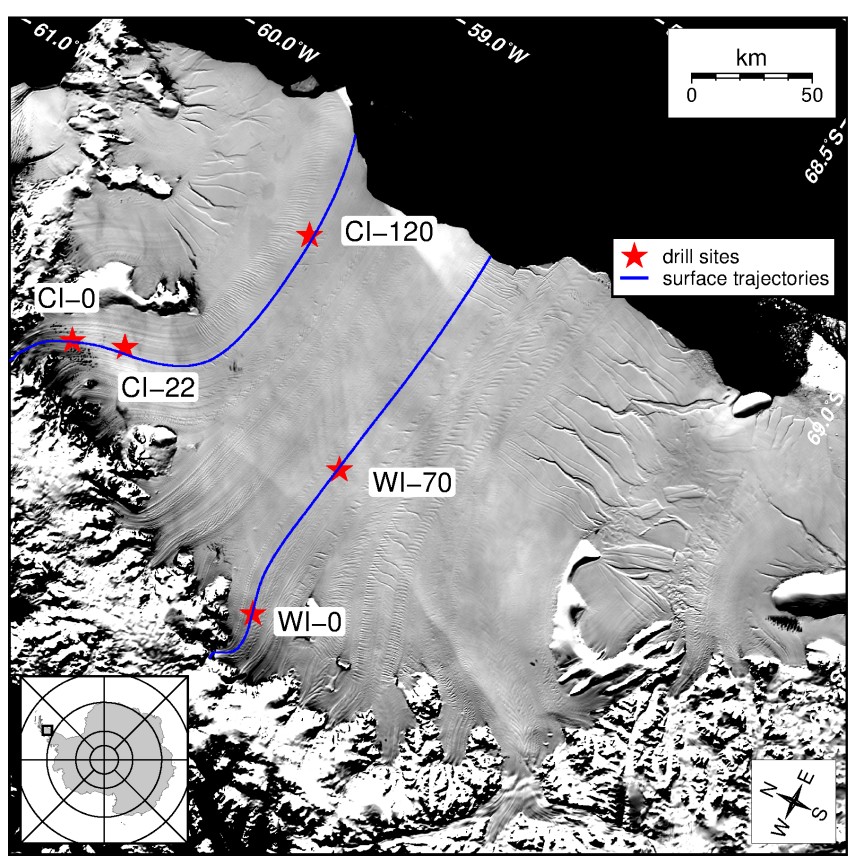

**Figure 1.** Mosaic of Antarctica (MOA2009) image of Larsen C ice shelf (Scambos et al., 2007; Haran et al., 2014) with borehole locations and Cabinet and Whirlwind Inlet surface flow trajectories.





**Figure 2.** OPTV images, density profiles, unit classifcations and binary thresholding output for boreholes at a) CI-0, b) CI-22 and c) CI-120. Figures reproduced from Ashmore et al. (2017).





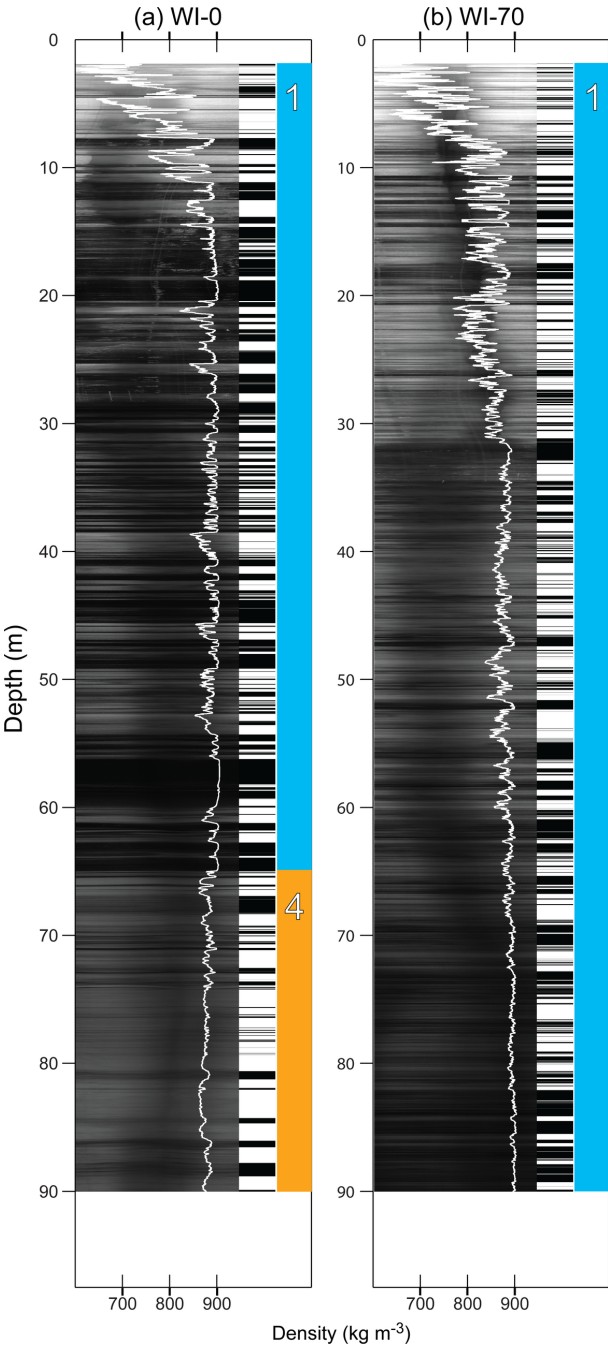

**Figure 3.** OPTV images, density profiles, unit classifcations and binary thresholding output for boreholes at a) WI-0 and b) WI-22. Figures reproduced from Ashmore et al. (2017).





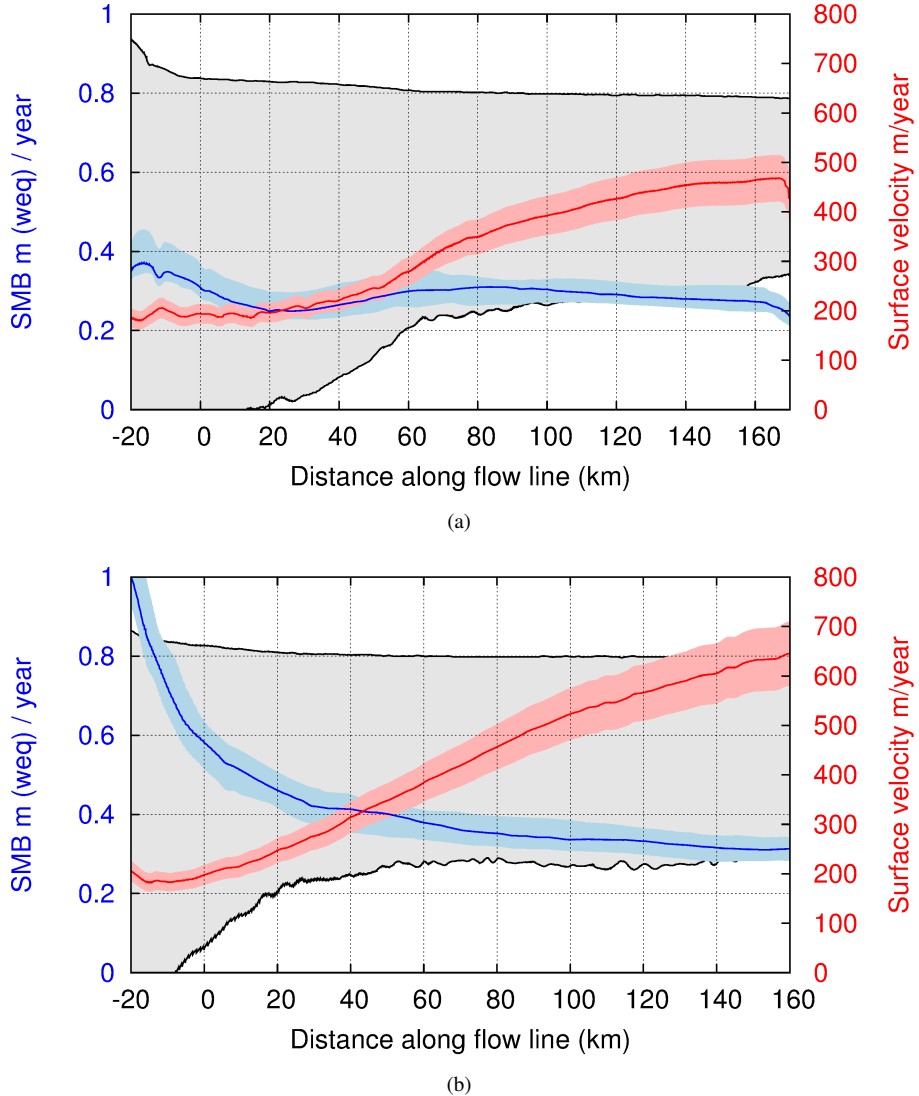

(a)

(b)

**Figure 4.** Along flow profiles of surface mass balance and velocity used for a) the Cabinet Inlet flowline and b) the Whirlwind Inlet flowline. the shaded boundaries represent the error estimates as described in the text.



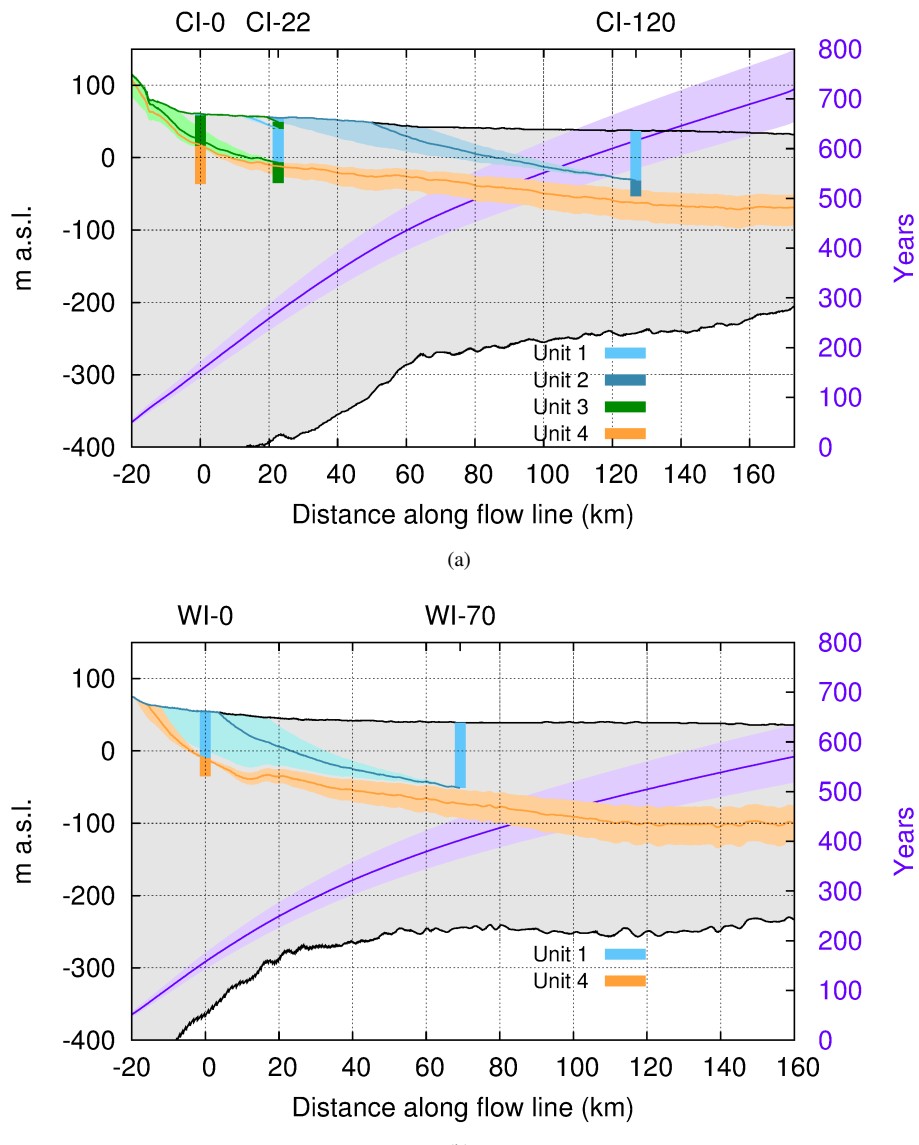

**Figure 5.** Flowlines initiated along trajectories from the grounding line towards the ice-shelf edge for a) Cabinet Inlet and b) Whirlwind Inlet. The time scale is from the beginning of the trajectory. The borehole ice units correspond to those described in Ashmore et al. (2017) and are described in the text. The shaded boundaries represent the error estimates as described in the text.





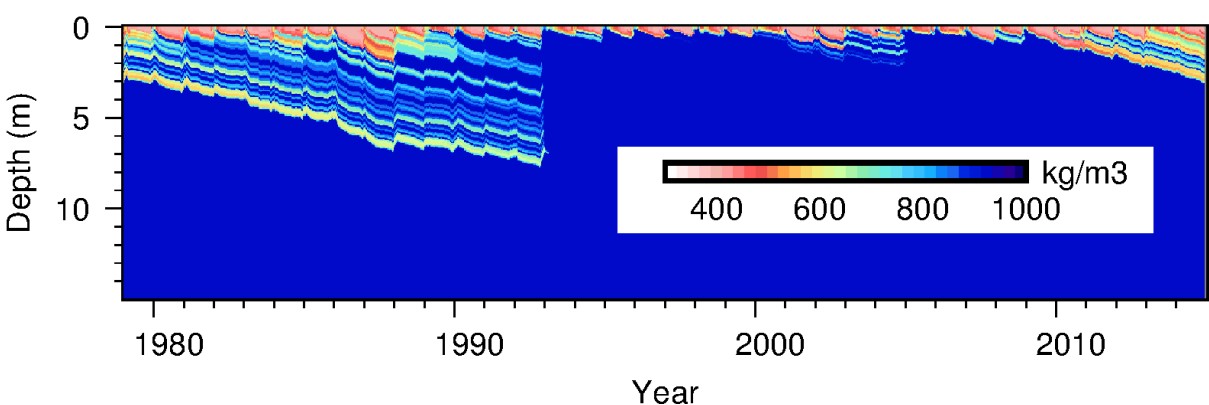

**Figure 6.** Time evolution of firn density as a function of depth for Cabinet Inlet, predicted by the firn density model.