# Peer review of "Centuries of intense surface melt on Larsen C Ice Shelf"

_The Cryosphere, 2017_

## Referee Comment (RC1) · Anonymous Referee #1 · 25 Jun 2017

Review of Bevan et al. A nice study of apparent thick ice layers in a set of ice cores, and the history of the climate and melt frequency on the Larsen C ice shelf that may be inferred from their depth and flow history.

I think the study needs minor revisions. It should not be published as it is (in my view) but a good and serious round of general improvement and attention will make it a good addition.

It seems like a lot of the content comes from earlier papers, so without a thorough look through all of the papers mentioned, it's a potential question as to what is new here. I presume the timing of warm periods and the connecting of the ice-layer indications along the flowline in time and vertical space are the main new contributions.

[Figure]

One fairly apparent question: in Greenland, in areas of moderate to high snow fall and abundant melting, water may percolate several meters, even tens of meters, before accumulating in massive soaked-firn layers called 'aquifers'. If this were happening on the Larsen C, what would this extensive vertical percolation do to your estimates of age and climate trends? Note that this implies that the melting could have occurred significantly downstream of the notional location, as well as later in time. — seems like maybe you are referring to this idea in P7L28-30. I think this can be addressed with a discussion in the Discussion. Refer to recent papers on the Greenland system by Koenig, and Forster.

P1L03 'known to be experience...' remove 'be' P1L09 remove currently – this area would always have been impacted by foehn events... well, for as long as there have been mountains and westerlies.... P1L10 'preconditioning of the ice' that would be 'snow'? P1L11 change to '.... that the modern period of melt ponding began.' P1L12 how deep? Can you give a range? And it would also be good to indicate if there was still a density anomaly relative to expected compaction – or was it an ice textural identification? P1L15 'Further south...' this sentence has more words than it needs. P1L18 '....we demonstrate that, even by the time...' Remove 'even by the time' you demonstrate that at the ice front, the ice shelf is comprised of 40 to 50% meteoric ice. P1L19 This last sentence comes rather 'out of the blue'. I suggest removing it, discussing what you want to say in the main text. You might also remove the preceding sentence as well – its just not clear where you are going here at the end. P2L04 Rott et al., 2002 primarily discusses the speed-up of Drygalski Glacier on the Larsen A, very little on the Larsen B breakup. Rack and Rott, 2004, Annals might be better here. P2L06 please add that cooler temperatures prevail over the ice shelf. Note that its unclear what the -9C limit really means. A summer limit of -2C or similar might make for a better link to the causes of retreat. Note that a -9C annual isotherm limit is unlikely to apply to any other region in Antarctica because of the different continentality of other regions (e.g. Ross, Fimbul) P2L08 Paolo et al., 2015 note a thickening in the most recent decade, when CryoSat-2 data is included. P3L06 – '...over the last 600

to 800 years..' explain this number. P7L18 '...are limited to the continental or basal accreted...' change to 'are only found within' or something similar. When I first read this here, and earlier in the paper, it seemed that you might be saying that the meteoric ice somehow prevented the basal crevasses from penetrating upward.

Table 1 – You should establish a reference year, such as (perhaps) 2015, and adjust your age ranges as needed with respect to that date. You may also wish to add a column of absolute ages on the C.E. scale. (In the future, others may want to relate your ages to layers deep within ice cores drilled in the 2020s or 2030s)

Figures 2 and 3 – please provide more explanation in the captions – what do the colors and numbers mean (binary classification), where exactly in the ice core is the evidence for increased surface ponding/melting episodes?

Figure 4 – what are the gray outlines in the two panels? Is that the ice shelf thickness? What are the units, which axis is active for the gray area? Ok, you have this in Figure 5 but perhaps the grey shaded areas should be removed from Figure 4, they are not used here.

Figure 5 – what is the lime-green section in the upper panel near the grounding line?

Another approach would be to merge Figure 4 and Figure 5 panels into one four-panel figure, and then refer the grey shaded area (which helps one track what is happening where in the vertical dimension of the shelf to the m.a.s.l. axis in the Figure 5 panels. This and your map would be the key figures.

It would be good to have a clearer Figure 2 and 3 as well, perhaps by lightening the gray-scale in the image, perhaps making it a clear-to-dark blue scale instead? With a yellow line and expanded scale (amplified 800 – 910 kg/m3 section) for the density. As it stands, Figure 2 and 3 are your main data, but are not helping the understanding much.

---

## Referee Comment (RC2) · E. Thomas (Referee) · 11 Sep 2017

The paper presents flowline and firn data density model data to interpret five 90m boreholes on Larsen C ice shelf. The study is timely, well written and clearly structured. I think the paper should be published following minor revisions.

There is a lot of reference to recently published papers and even a reproduction of a figure from a study published this year. I have not read these works but the author's state that Ashmore et al., 2017 concluded that spatial melt has been ongoing for decades to centuries. Perhaps the authors could make it clear what is new about this study (ice flow modelling? dating melt events?), demonstrating that this study builds on existing research but contains novel insights.

One area that could be improved is relating the ages of these melt events to the wider climate of the region. You mention instrumental evidence for warming beginning in the 1950s, but there is ice core evidence from the central and southern Antarctic Peinsula that this is part of a longer 20th century trend (eg Bruce Plateau and Gomez ice cores). In addition, the Ferrigno ice core revealed warming trends during the mid 18th and 19th centuries that would support your findings for melt events during those periods.

Relating to this, there is growing evidence that SMB on the AP has been changing dramatically during the 20th century. Admittedly the majority of the ice core records are from the western side of the Peninsula, but the snow accumulation records here are strongly influenced by changes in westerly wind strength (eg SAM), which is driving changes in fohn winds and impacting melt on Larsen C. My query therefore is has the snow accumulation on the eastern side of the AP remained stable during the past 300 years? And if not, how would that influence the flowline models and age estimates? Could this explain some of the discrepancies you mention (page 8)?

Technical corrections: Abstract – "experience", change to "experiencing" "..the boreholes sample ice that...." consider rewording? Page2, ln 26 – duplication "in which" Page3, ln 18 – "additional", unnecessary wording Page 4, - title capitalisation "Flowline model" Page 5, version of RACMO? 2.3? Perhaps define eg "....the Regional Atmospheric Climate Model (RACMO2.3)." Page 5, ln 29 the estimates of 870 and 588 years are from this study? Page 19 delete "along"

---

## Author Comment (AC1) · 28 Sep 2017

Review of Bevan et al. A nice study of apparent thick ice layers in a set of ice cores, and the history of the climate and melt frequency on the Larsen C ice shelf that may be inferred from their depth and flow history. I think the study needs minor revisions. It should not be published as it is (in my view) but a good and serious round of general improvement and attention will make it a good addition.

It seems like a lot of the content comes from earlier papers, so without a thorough look through all of the papers mentioned, it's a potential question as to what is new here. I presume the timing of warm periods and the connecting of the ice-layer indications along the flowline in time and vertical space are the main new contributions.

*Point accepted. We have tried to make this clearer in the introduction by stating that the description of the ice units given is a summary of Hubbard et al. (2016) and Ashmore et al. (2017). We have also changed the concluding paragraph of the introduction to…*

> *'Ashmore et al. (2017) concluded that the significant quantities of refrozen ice within the boreholes suggests that intense melt is spatially pervasive and has been ongoing on LCIS for decades or even centuries. In this study we use a flowline model to investigate where and when various units of melt-affected ice observed within the Cabinet Inlet and four other boreholes originated, relate the origins to past local climate, and estimate how much of the ice shelf is likely to be affected.'*

One fairly apparent question: in Greenland, in areas of moderate to high snow fall and abundant melting, water may percolate several meters, even tens of meters, before accumulating in massive soaked-firn layers called 'aquifers'. If this were happening on the Larsen C, what would this extensive vertical percolation do to your estimates of age and climate trends? Note that this implies that the melting could have occurred significantly downstream of the notional location, as well as later in time. — seems like maybe you are referring to this idea in P7L28-30. I think this can be addressed with a discussion in the Discussion. Refer to recent papers on the Greenland system by Koenig, and Forster.

*Lines 28-34 on page 8 already expanded further on this idea. We have added a line referencing Koenig et al. (2017) so that this paragraph now reads…*

> *'Lateral influx of meltwater is also a real possibility in this locality. By analogy with observations within the percolation zone of the Greenland Ice Sheet (Harper et al. 2012, Machguth et al. 2016), the formation of spatially discontinuous impermeable near-surface layers of ice following melt-refreeze events, would facilitate horizontal flow of meltwater along and across the troughs in which the melt ponds form. Vertical infiltration at the boundaries of the ice barriers would*

*result in a horizontally heterogeneous distribution of U3 type bodies. The importance of horizontal liquid water transport and its dependence on surface slope on the Greenland Ice Sheet is emphasised by Forster et al. (2014). As described in Hubbard et al. (2016), borehole CI-0 was drilled into a melt-pond trough which might be expected to contain a local concentration of infiltration ice.'*

P1L03 'known to be experience…' remove 'be'

*Changed to 'known to be experiencing…'*

P1L09 remove currently – this area would always have been impacted by foehn events… well, for as long as there have been mountains and westerlies.

*Agreed, done.*

P1L10 'preconditioning of the ice' that would be 'snow'?

*Thank you, changed.*

P1L11 change to '… that the modern period of melt ponding began.'

*Thank you, changed*

P1L12 how deep? Can you give a range? And it would also be good to indicate if there was still a density anomaly relative to expected compaction – or was it an ice textural identification?

*Depths added. We have made it clearer that these units are anomalous on the basis of density by stating 'which have densities exceeding those expected under normal compaction metamorphism'.*

P1L15 'Further south: …' this sentence has more words than it needs.

*Changed from 'Further south on the shelf…' to 'Further south…'*

P1L18 '…we demonstrate that, even by the time…' Remove 'even by the time' you demonstrate that at the ice front, the ice shelf is comprised of 40 to 50% meteoric ice.

*Done.*

P1L19 This last sentence comes rather 'out of the blue'. I suggest removing it, discussing what you want to say in the main text. You might also remove the preceding sentence as well – it's just not clear where you are going here at the end.

*Agreed, this part is not necessary in the abstract and has been removed.*

P2L04 Rott et al., 2002 primarily discusses the speed-up of Drygalski Glacier on the Larsen A, very little on the Larsen B breakup. Rack and Rott, 2004, Annals might be better here.

*Agreed and changed, thank you.*

P2L06 please add that cooler temperatures prevail over the ice shelf. Note that It's unclear what the -9C limit really means. A summer limit of -2C or similar might make for a better link to the causes of retreat. Note that a -9C annual isotherm limit is unlikely to apply to any other region in Antarctica because of the different continentality of other regions (e.g. Ross, Fimbul)

*We have added the phrase 'with cooler mean annual temperatures on the shelf'. Although the –2C might be a better limit, we have not analysed this, but rather adopted the -9C annual isotherm as the limit identified by Morris and Vaughan (2003). We have added 'Antarctic Peninsula ice-shelf viability.'*

P2L08 Paolo et al., 2015 note a thickening in the most recent decade, when CryoSat-2 data is included.

*The Paolo et al. (2015) paper cited in the manuscript shows a thickening only in the very south-east corner of the shelf. However, this paper does not include cryostat-2 data. As far as we are aware, the recent widespread thickening is not yet published so we have been unable to refer to it.*

P3L06 – '…over the last 600 to 800 years..' explain this number.

*From following the full trajectories we know that this is how long it takes ice to travel from the region where ponding occurs to the edge of the shelf. To make this clearer, we have removed this phrase so that the sentence reads*

> *'The effect of foehn-induced melting and ponding on the englacial properties of the ice shelf downstream depends on the history of surface melt, and consequently how far along-flow its temperature and density legacy has been advected.'*

P7L18 '…are limited to the continental or basal accreted…' change to 'are only found within' or something similar. When I first read this here, and earlier in the paper, it seemed that you might be saying that the meteoric ice somehow prevented the basal crevasses from penetrating upward.

*Done.*

Table 1 – You should establish a reference year, such as (perhaps) 2015, and adjust your age ranges as needed with respect to that date. You may also wish to add a column of absolute ages on the C.E. scale. (In the future, others may want to relate your ages to layers deep within ice cores drilled in the 2020s or 2030s).

*Good point, we have changed ages to CE dates in Table 1 and throughout the text and added*

> *'We convert the ages to dates by taking 2015, the year of the latest borehole observations, to be year 0.'*

Figures 2 and 3 – please provide more explanation in the captions – what do the colors and numbers mean (binary classification), where exactly in the ice core is the evidence for increased surface ponding/melting episodes?

*We have added a lot more detail to the caption of Fig. 2, it now reads…*

> *'OPTV images, density profiles, unit classifications and binary thresholding output for boreholes at a) CI-0, b) CI-22 and c) CI-120. The grey shading represents the recorded luminosity of the ice, and the white profile the inferred density using the x-axis scale. The coloured panels are the different ice type units referred to in the text. The black strips in the bar-code like panels show the presence of refrozen ice determined via a binary thresholding analysis which was deemed to perform poorly for Unit 3 ice (Ashmore et al., 2017). Figures reproduced from Ashmore et al. (2017) where the methods are described in more detail.'*

Figure 4 – what are the gray outlines in the two panels? Is that the ice shelf thickness? What are the units, which axis is active for the gray area? Ok, you have this in Figure 5 but perhaps the grey shaded areas should be removed from Figure 4, they are not used here.

*We have kept the ice shelf outlines in Fig. 4 to aid locating the profiles with respect to the ice shelf. However, we have labelled the figure to make this clear and used the caption to refer to the scales in Fig. 5.*

Figure 5 – what is the lime-green section in the upper panel near the grounding line?

*This is the shaded uncertainty region referred to in the caption. We have muted the colour slightly to match the colour of the unit 3 ice more closely.*

Another approach would be to merge Figure 4 and Figure 5 panels into one four-panel figure, and then refer the grey shaded area (which helps one track what is happening where in the vertical dimension of the shelf to the m.a.s.l. axis in the Figure 5 panels.

*See above. We would prefer to keep the figures separate so that Fig. 5 panels can remain full page width.*

This and your map would be the key figures. It would be good to have a clearer Figure 2 and 3 as well, perhaps by lightening the gray-scale in the image, perhaps making it a clear-to-dark blue scale instead? With a yellow line and expanded scale (amplified 800 – 910 kg/m3 section) for the density.

*These figures, are reproductions of figures published in Ashmore et al. (2017). The grey scale is 'true' colour of the reflected light in the borehole and we would prefer to keep it as it is. However, we have made the density trace yellow so that it stands out more. The scale needs to remain as is in order to include the lower densities seen near the surface.*

As it stands, Figure 2 and 3 are your main data, but are not helping the understanding much.

*We hope that the redrawn figure and reworded caption help to improve the understanding.*

*Please note that following the calving event in July this year we have made a few modifications to the introduction and now P2L10 reads*

> *'In July 2017 a rift, which began propagating from the south in 2014 (Jansen et al., 2015; Borstad et al., 2017}, caused ~10% of the ice shelf area to break away (Hogg and Gudmundsson, 2017).'*

---

## Author Comment (AC2) · 28 Sep 2017

The paper presents flowline and firn data density model data to interpret five 90m boreholes on Larsen C ice shelf. The study is timely, well written and clearly structured.

I think the paper should be published following minor revisions.

There is a lot of reference to recently published papers and even a reproduction of a figure from a study published this year. I have not read these works but the author's state that Ashmore et al., 2017 concluded that spatial melt has been ongoing for decades to centuries. Perhaps the authors could make it clear what is new about this study (ice flow modelling? dating melt events?), demonstrating that this study builds on existing research but contains novel insights.

*Point accepted and (as in our reply to Reviewer 1) we have tried to make this clearer in the introduction by stating that the description of the ice units given is a summary of Hubbard et al. (2016) and Ashmore et al. (2017). We have also changed the concluding paragraph of the introduction to*

> *'Ashmore et al. (2017) concluded that the significant quantities of refrozen ice within the boreholes suggests that intense melt is spatially pervasive and has been ongoing on LCIS for decades or even centuries. In this study we use a flowline model to investigate where and when various units of melt-affected ice observed within the Cabinet Inlet and four other boreholes originated, relate the origins to past local climate, and estimate how much of the ice shelf is likely to be affected.'*

One area that could be improved is relating the ages of these melt events to the wider climate of the region. You mention instrumental evidence for warming beginning in the 1950s, but there is ice core evidence from the central and southern Antarctic Peninsula that this is part of a longer 20th century trend (eg Bruce Plateau and Gomez ice cores). In addition, the Ferrigno ice core revealed warming trends during the mid 18th and 19[th] centuries that would support your findings for melt events during those periods.

*We had mentioned the Bruce Plateau core but have now added reference to the Gomez and also Ferrigno ice cores in discussing the 20[th] century warming. We have also added the following to the discussion of the 18[th] century warming. Thank you for the suggestion.*

> *'Although the Ferrigno ice core indicated a warming in the second rather than first half of the 18th century (Thomas et al., 2013) it reveals, along with the JRI core, that the AP region has*

*experienced a decadal-scale variability in air temperature over the past 300 to 1000 years of a similar magnitude to the 20th century warming.'*

Relating to this, there is growing evidence that SMB on the AP has been changing dramatically during the 20th century. Admittedly the majority of the ice core records are from the western side of the Peninsula, but the snow accumulation records here are strongly influenced by changes in westerly wind strength (eg SAM), which is driving changes in fohn winds and impacting melt on Larsen C. My query therefore is has the snow accumulation on the eastern side of the AP remained stable during the past 300 years? And if not, how would that influence the flowline models and age estimates? Could this explain some of the discrepancies you mention (page 8)?

*This is an interesting point but the discrepancies regarding dates in the inlet regions are more likely to result from difficulties in modelling accumulation downstream of steep topography (as well as not having accounted for lateral meltwater influx and vertical percolation). Since submitting the manuscript an observationally constrained improved reconstructed SMB field has become available (Kuipers Munneke et al., 2017). Although this increases SMB estimates over most of each flowline we have chosen not to use it as explained in the paragraph below which has been added to Section 2.3 Surface mass balance.*

> *Since this paper was reviewed for publication, an observationally constrained improved reconstruction of LCIS 1979–2015 mean SMB has become available (Kuipers Munneke et al., 2017) which exhibits values ~10% higher than our upper uncertainty bound along each trajectory. Despite this apparently improved dataset being available prior to final publication, we have not updated our analysis because the new dataset will not significantly impact on our results, discussion or conclusions, and is probably not an improvement for the context in which we are using it. In short, we are necessarily approximating a long chronology (more than 300 years) of values using a relatively short contemporary SMB field. In this context, the new reconstruction will not offer an improvement especially as there is some evidence for accumulation rates having increased by over 10% in Antarctic coastal regions since the 1960s (Frezzotti et al., 2013), and therefore the lower SMB values from our original dataset are probably a better representation of the longer-term estimate.*

Technical corrections: Abstract – "experience", change to "experiencing"

*Thank you, changed.*

"..the boreholes sample ice that...." consider rewording?

*We would like to keep the wording as it is.*

Page2, ln 26 – duplication "in which"

*Thank you, changed.*

Page3, ln 18 – "additional", unnecessary wording

*We have kept in 'additional' although this paragraph has been reworded slightly in addressing reviewers' suggestions that we make clearer what is existing research and what is new.*

Page 4, - title capitalisation "Flowline model"

*Thank you, changed.*

Page 5, version of RACMO? 2.3? Perhaps define eg "….the Regional Atmospheric Climate Model (RACMO2.3)."

*Done.*

Page 5, ln 29 the estimates of 870 and 588 years are from this study?

*Yes, have added '…we calculate…' to make it clear.*

Page 19 delete "along"

*'along' is needed but we have hyphenated 'along-flow profiles'.*

*Please note that following the calving event in July this year we have now changed P2L10 to*

> *'In July 2017 a rift, which began propagating from the south in 2014 (Jansen et al., 2015; Borstad et al., 2017}, caused ~10% of the ice shelf area to break away (Hogg and Gudmundsson, 2017).'*